# MaCts1, an Endochitinase, Is Involved in Conidial Germination, Conidial Yield, Stress Tolerances and Microcycle Conidiation in *Metarhizium acridum*

**DOI:** 10.3390/biology11121730

**Published:** 2022-11-29

**Authors:** Yuneng Zou, Chan Li, Shuqin Wang, Yuxian Xia, Kai Jin

**Affiliations:** 1Genetic Engineering Research Center, School of Life Sciences, Chongqing University, Chongqing 401331, China; 2Chongqing Engineering Research Center for Fungal Insecticide, Chongqing 401331, China; 3Key Laboratory of Gene Function and Regulation Technologies Under Chongqing Municipal Education Commission, Chongqing 401331, China

**Keywords:** entomopathogenic fungus, Cts1, endochitinase, microcycle conidiation, MOR/RAM pathway

## Abstract

**Simple Summary:**

Entomopathogenic fungi are promising biocontrol agents. Microcycle conidiation has shown great potential in enhancing the conidial yield and quality of entomopathogenic fungi. Elucidating the regulatory mechanisms underlying the induction of microcycle conidiation will be helpful for genetic improvement of entomopathogenic fungi. This work focused on the connection of an endochitinase, MaCts1, to the biocontrol potential of the entomopathogenic fungus *Metarhizium acridum*. *MaCts1*, an endochitinase gene of *M. acridum*, was shown to be involved in conidial germination, conidial yield and fungal resistance to UV-B irradiation and heat-shock. Interestingly, disruption of *MaCts1* led to maintenance of typical conidiation on the microcycle conidiation induction medium, SYA, which may not be dependent on the MOR/RAM pathway. This work provided insight into the regulatory mechanisms governing the shift to microcycle conidiation in *M. acridum*.

**Abstract:**

Entomopathogenic fungi are promising biocontrol agents of insect-mediated crop damage. Microcycle conidiation has shown great potential in enhancing the conidial yield and quality of entomopathogenic fungi. Homologs of Cts1, an endochitinase of *Saccharomyces cerevisiae*, participate in cell separation in several fungal spp. and may contribute to the morphological differences that occur during the shift to microcycle conidiation. However, the precise functions of *Cts1* in entomopathogenic fungi remain unclear. Herein, the endochitinase gene, *MaCts1*, was characterized in the model entomopathogen, *Metarhizium acridum*. A loss of function line for *MaCts1* led to a delay of 1 h in the median germination time, a 28% reduction in conidial yield and significant defects in fungal resistances to UV-irradiation (18%) and heat-shock (15%), while fungal tolerances to cell wall stressors, oxidative and hyperosmotic stresses and virulence remained unchanged. The *MaCts1*-disruption strain displayed typical conidiation on the microcycle conidiation induction medium, SYA. In contrast, deletion of key genes in the morphogenesis-related NDR kinase network (MOR pathway)/regulation of Ace2 and morphogenesis (RAM pathway) did not affect the SYA-induction of microcycle conidiation. This indicates that *MaCts1* makes contributions to the microcycle conidiation, which may not be dependent on the MOR/RAM pathway in *M. acridum*.

## 1. Introduction

Insect pathogenic fungi are considered as promising biocontrol agents due to their lack of toxicity to humans and animals, and their low likelihood to cause host resistance [1,2]. Infection of the insect host involves conidial adherence to the insect epicuticle, followed by conidial germination, and the development of an infection structure (appressorium), which can penetrate the host epicuticle to utilize the nutrition in the hemocoel, leading to insect death. The fungus then penetrates throughout the host cadaver for the large-scale production of conidia for repetition of the cycle [3,4]. Usually, aerial conidia are not only the infection units of entomopathogenic fungi, but are also the main active ingredients of mycopesticides [5,6,7]. Thus, the conidial yield and quality directly determine the production cost and the efficiency of these mycoinsecticides. There are two different conidiation patterns in most filamentous fungi [8,9]. In typical conidiation, the fungus conducts a period of mycelial growth before conidiation [9,10,11]. Environmentally limiting conditions can trigger microcycle conidiation, where the conidia are directly produced from germinated conidia without any hyphal growth [8,12]. Microcycle conidia can be more tolerant to environmental conditions than typical conidia, such as the higher trehalose in microcycle conidia of *M. acridum*, which has been reported to confer higher tolerance to heat stress [13]. Thereby, the identification of further genes involved in microcycle conidiation, conidial yield and quality would be beneficial for the genetic improvement of entomopathogenic fungi as biocontrol agents.

Fungal cell walls play critical roles in maintaining cell morphology and resisting environmental stress [14,15]. Chitin is a conserved component of fungal cell walls and contributes to defense and morphological maintenance [16]. The synthesis and decomposition of chitin in fungi involves exo- and endo-chitinases belonging to the CAZy (Carbohydrate-Active enZYmes) family GH18 (glycoside hydrolases) [17,18,19]. Exochitinases degrade chitin chains from their non-reducing or reducing ends, while endochitinases hydrolyze glycosidic randomly bonds in amorphous zones within the chitin chain, contributing to nutrition, cellular antagonism, polar growth and cell separation [19,20]. Cts1, a typical endochitinase, can translocate into the fragmentation zone to degrade chitin of the primary septum for detaching cells in the late stage of cytokinesis in *Saccharomyces cerevisiae* [21]. Deletion of *Cts1* leads to a cell separation defect and the promotion of pseudohyphal growth, resulting in cell clumping and altered colony morphology [22,23]. In *Aspergillus nidulans*, the disruption of the *Cts1* homolog, *chiA*, impairs spore germination and hyphal growth, leading to morphological changes due to cell wall damage [24]. In *Candida albicans*, the loss of the *Cts1* homolog, *CHT3*, leads to defective cytokinesis between mother and daughter cells, with alteration in both axes of polarity, producing uneven colony edges [25]. However, the precise functions of *Cts1* homologs in entomopathogenic fungi remain unclear.

In fungi, the regulation of morphology and cell separation occurs via two similar pathways: the morphogenesis-related NDR kinase network (MOR) and regulation of the Ace2 and morphogenesis (RAM) pathway [26,27,28,29]. Both pathways contain similar components, including *Hym1*, *Kic1*, *Tao3*, *Cbk1* and *Mob2*, but the RAM pathway additionally contains the zinc finger transcription factor Ace2 [30,31]. Thus, for clarity and completeness, we used the MOR/RAM signaling network to represent them throughout this paper. The MOR/RAM pathways have been shown to regulate the expression of *Cts1* during cell separation in several fungi [22,23].

Herein, the *M. acridum Cts1* homolog, *MaCts1*, has been characterized. The disruption of *MaCts1* resulted in delayed conidial germination, reduced conidial yield and impaired fungal resistances to UV-irradiation and heat-shock, while the fungal sensitivities to cell wall stressors, oxidative and hyperosmotic stresses and virulence remained unaffected. Interestingly, *MaCts1* inactivation resulted in altered induction of microcycle conidiation, so that typical conidiation was obtained on the microcycle conidiation induction medium, SYA. In contrast, deletions of key genes in the MOR/RAM signaling network, respectively, such as *MaHym1*, *MaKic1*, *MaCbk1*, *MaMob2* and *MaTao3*, and a transcription factor gene, *MaAce2*, did not change the microcycle conidiation in *M. acridum*. Taken together, these results indicate that *MaCts1* contributes to microcycle conidiation, the resistance to UV-irradiation and heat-shock in *M. acridum***.**

## 2. Materials and Methods

### 2.1. Strains and Culture Conditions

The fungal strain *M. acridum* CQMa102 (WT) was cultivated on 1/4 Sabouraud’s dextrose agar plus 1% yeast extracts (1/4 SDAY) medium as previously described [32]. *Escherichia coli* DH5α (TransGen Biotech, Beijing, China) was utilized for routine DNA manipulations and grown on Luria-Bertani (LB) medium with 50 μg/mL kanamycin (Dingguo Biotechnology, Beijing, China) at 37 °C. *Agrobacterium tumefaciens* AGL-1 (Weidi Biotech, Shanghai, China) was used for fungal transformations and grown on LB medium with 50 μg/mL kanamycin at 28 °C.

### 2.2. Gene Deletion and Complementation

The disruption vectors (for *MaCts1* (MAC-08492), *Mahym1* (MAC-08749), *MaKic1* (MAC-05921), *MaCbk1* (MAC-04187), *MaMob2* (MAC-02988), *MaAce2* (MAC-04654) and *MaTao3* (MAC-01346)) were constructed using the 5′- and 3′-flanking fragments (~1000 bp) of each gene cloned into the two sides of a *bar* cassette of the pK2-PB vector [33]. The final vectors were transformed into CQMa102 using *Agrobacterium*-mediated method [33]. Czapek–Dox medium containing 500 µg/mL glufosinate ammonium (Sigma, St. Louis, MO, USA) was used to select the putative disruption mutants for validation by PCR or Southern blotting. To rescue the *MaCts1* in the *MaCts1*-disruption mutant (Δ*MaCts1*), the promoter and coding regions of the *MaCts1* were cloned into the pK2-Sur vector [33] to form the *MaCts1* rescuing vector, pK2-Sur-*MaCts1*, which was inserted into Δ*MaCts1* via *Agrobacterium*-mediated method [34]. The complemented transformants (CP) of *MaCts1* were screened on Czapek–Dox medium containing 20 µg/mL chlorimuron ethyl (Sigma, Bellefonte, PA, USA). To confirm the *MaCts1* deletion and complementation strains, Southern blotting was carried out as previously described [35]. Genomic DNA (~5 µg/sample) was digested with *Stu*I/*Eco*RI. All primers used are listed in Appendix A.

### 2.3. Phenotypic Analyses

Conidial germination assays were conducted as previously described [36]. In brief, conidial suspensions of fungal strains were prepared with sterile 0.05% Tween-80, followed by filtering the suspension through four layers of sterile lens paper to remove mycelia. The concentrations of conidial suspension were determined using a hemocytometer. Aliquots of 50 µL conidial suspension (1 × 10^7^ conidia/mL) were inoculated onto 1/4 SDAY plates at 28 °C. Germination rates of the WT, Δ*MaCts1*, CP strains were assessed every 2 h until 95% of conidia were germinated. Conidial yield was assayed as previously described [37]. Briefly, 2 µL of conidial suspensions (1 × 10^6^ conidia/mL) from the WT, Δ*MaCts1*, CP strains were dropped into each well of the 24-cell plates, which contained 1 mL of l/4 SDAY per well, and incubated at 28 °C. Conidia were collected every 3 days from wells by washing with 1 mL of 0.1% Tween-80 for counting by hemocytometer. Fungal resistances to UV-irradiation and heat-shock were assayed as previously described [33]. Briefly, aliquots of 50-μL conidial suspensions (1 × 10^7^ conidia/mL) from the WT, Δ*MaCts1*, CP strains were inoculated onto 1/4 SDAY plates, and then treated by either UV-B irradiation (1350 mW/m^−2^) for 0, 1.5, 3.0, 4.5 and 6.0 h, or incubated at 44.5 °C for 0, 3, 6, 9 and 12 h. In each case, the plates were incubated at 28 °C for 20 h and their conidial germination rates were calculated by hemocytometer as described above. Fungal tolerances to hyperosmotic and oxidative stresses were determined by spot assays on 1/4 SDAY plates with 1 mol/L NaCl, 1 mol/L sorbitol (SOR) and 0.07% *w*/*v* Menadione. Fungal sensitivities to cell wall stressors were determined by spot assays on 1/4 SDAY plates with 0.01% *w*/*v* sodium dodecyl sulfate (SDS), 50 μg/mL calcofluor white (CFW) and 500 μg/mL Congo red (CR). Conidial suspensions (2 μL of 1 × 10^6^ conidia/mL) from the WT, Δ*MaCts1*, CP strains were spotted on 1/4 SDAY alone or supplemented with stressors, then cultivated for 6 days at 28 °C before photographing fungal colonies. Triplicate experiments were carried out.

### 2.4. Fungal Pathogenicity Assays

Conidial suspensions in paraffin oil (5 µL of 1 × 10^7^ conidia/mL) of the WT, Δ*MaCts1*, CP strains were topically inoculated onto fifth-instar stage nymphs of *Locusta migratoria manilensis* (Meyen) [33]. The surviving number of insects was recorded every 12 h to calculate the half lethality time (LT_50_) using the Data Processing System program [38], which was used as a measure of the virulence of the fungal strains.

### 2.5. Microscopic Observation of the Conidiation

To observe conidiation processes of different fungal strains, conidial suspensions (100 µL of 1 × 10^7^ conidia/mL) were inoculated on 1/4 SDAY plates and the microcycle conidiation medium (SYA, 0.5% yeast extract, 3% sucrose, 0.05% MgSO_4_, 0.001% MnSO_4_, 0.05% KCl, 0.3% NaNO_3_, 0.1% KH_2_PO_4_, 0.001% FeSO_4_ and 2% agar, *w*/*v*), followed by incubation at 28 °C. After 16, 20, 24, 28 and 36 h of cultivations, approximately 1 cm^2^ was cut for observation using a microscope (Motic, Guangzhou, China). Hyphal samples were stained with 10 µL CFW (50 μg/mL) for 30 min after 18, 24 and 30 h of cultivation to visualize the mycelial septa with a fluorescent microscope (Nikon Eclipse Ci-E, Tokyo, Japan).

### 2.6. Data Analysis

In this study, the data from each time point (hours or days) were considered as a single replicate. At first, the normality and uniformity of the original data were analyzed using the Shapiro–Wilk test. After confirmation, the means were compared by Tukey’s test. Non-normally distributed, data were further analyzed by a Mann–Whitney U test [38,39]. All the datasets in this study were normally distributed, and one-way analysis of variance (ANOVA) was applied to access the phenotypic estimate, including conidial germination rates with different treatments (UV-B, Heat-shock), conidial yield, locust survival and cell length in the conidiation process of WT, Δ*MaCts1*, CP. Datasets were analyzed with the SPSS 18.0 program (SPSS Inc, Chicago, IL, USA). All experiments were triplicated.

## 3. Results

### 3.1. Features of MaCts1

The *Cts1* homolog, *MaCts1* of *M. acridum*, was retrieved from the CQMa102 genome. The *MaCts1* has a 1401 bp open reading frame, which encodes a predicted 392 amino acids protein (41.27 kDa). Domain prediction using the web resource SMART (http://smart.embl.de/ (accessed on 14 April 2022) showed its content of a glycosyl hydrolase family 18 domain (Glyco_18) and a C-terminal cellulose binding domain (CBM_1) existed in the deduced MaCts1 (Figure 1A). A phylogenetic analysis of MaCts1 and other fungal Cts1 homologs showed that it was more closely related to *M. robertsii*, *M. rileyi* and *B. bassiana*, with sequence identities of 52.05%, 45.98% and 37.24%, respectively (Figure 1B).

### 3.2. MaCts1 Is Involved in Conidial Germination and Influences Conidial Yield but Not Virulence

Southern blotting showed that *MaCts1* were successfully disrupted and rescued (Appendix A). Assessment of the conidial germination rates of the WT, Δ*MaCts1* and CP strains indicated that Δ*MaCts1* exhibited a significant delay relative to the control strains (WT and CP). The calculations of GT_50_ showed that Δ*MaCts1* (6.43 ± 0.31 h) was significantly longer than the WT (5.10 ± 0.24 h) and CP (5.34 ± 0.21 h) strains (*p* < 0.05; Figure 2A,B). The yield of conidia from Δ*MaCts1* grown on 1/4 SDAY media for 15 days at 28 °C was remarkably decreased by 28% (Figure 2C). Microscopic observation indicated that the control strains had produced conidiophores at 24 h of incubation, whereas conidiophores emerged in Δ*MaCts1* only after 36 h (Figure 2D). Insect bioassays via topical inoculation showed that *MaCts1* made no significant contribution to fungal virulence (Figure 2E,F).

### 3.3. Disruption of MaCts1 Impairs Fungal Resistances to UV-B Irradiation and Heat-Shock

In general, the conidial germination rates decreased with longer heat-shock or UV-B irradiation treatments in all strains (Figure 3A,C). However, the Δ*MaCts1* strain displayed significantly higher sensitivity to UV-B irradiation and heat-shock than the control strains. After 3-h UV-B irradiation, ~50% of conidia from Δ*MaCts1* germinated, compared to ~75% of conidia from the control strains (*p* < 0.01; Figure 3A). The half germination time of inhibition (IT_50_) of Δ*MaCts1* was significantly decreased by 18% (*p* < 0.01; Figure 3B). After 6 h of heat-shock, only ~36.0% conidia of Δ*MaCts1* germinated, compared to more than 50% conidia from the control strains (*p* < 0.01; Figure 3C), with a significant reduction in the IT_50_ of Δ*MaCts1* of 15% (*p* < 0.01; Figure 3D). In addition, fungal sensitivities to oxygen stress (menadione), cell wall stress (CR, CFW, SDS) and osmotic stress (SOR, NaCl) were assessed and measured by morphology and diameters of fungal colonies on 1/4 SDAY medium. The results indicated that the colony morphology of Δ*MaCts1* showed no obvious differences to that of the control strains (Figure 3E).

### 3.4. MaCts1 Makes Contributions to the Microcycle Conidiation

To understand the roles of *MaCts1* during the microcycle conidiation of *M. acridum*, the conidiation process of the WT, Δ*MaCts1* and CP strains grown on SYA medium was microscopically observed. The result was that microcycle conidiation was induced in the control strains, whereas Δ*MaCts1* exhibited the pattern of typical conidiation (Figure 4A). Disruption of *MaCts1* also occurred with a significant increase in the apical and sub-apical cell lengths on SYA medium (*p* < 0.001; Figure 4B–D).

The MOR/RAM pathway plays an important regulatory role in fungal morphogenesis [26,28], and has been reported to regulate the expression of *Cts1* during cell separation in several fungi [22,23]. Thus, we conducted BLASTp sequence similarity searches based on the full protein sequences of MOR/RAM components from *S. cerevisiae*, *S. pombe* [40] and *U. maydis* [21] to identify the putatively conserved MOR/RAM components from the *M. acridum* genome (Appendix A). The key components of the MOR/RAM, including two essential serine/threonine protein kinase genes, *Cbk1* and *Kic1*, three associated protein genes, *Hym1*, *Mob2* and *Tao3*, and an additional transcription factor gene, *Ace2*, were identified and disrupted in *M. acridum*. All the disrupted mutants were successfully confirmed by PCR (Appendix A) and, in all, microcycle conidiation was successfully induced on SYA medium, while only Δ*MaCts1* exhibited the typical conidiation pattern (Figure 5). These results indicate that *MaCts1* contributes to the microcycle conidiation, but that the regulation of *MaCts1* may be independent on the MOR/RAM pathway.

## 4. Discussion

Chitin is the main polysaccharide component of fungal cell walls and is critical for fungal cell morphogenesis [16,41]. Chitin is degraded by endo- and exochitinases to provide nutrition to fungal cells and determine fungal morphology [40,41,42]. In this study, *MaCtst1*, a gene encoding the *M. acridum* homolog of the Cts1 endochitinase, was characterized. Deletion of *MaCts1* led to severe defects in conidial germination and conidial yields, with decreased resistances to UV-B irradiation and heat-shock. Interestingly, we found *MaCts1* contributes to the microcycle conidiation of *M. acridum*, as shown by the inability of the Δ*MaCts1* strain to produce microcycle conidia on SYA media and the restoration of this ability in CP strains. In *S. cerevisiae* and other fungi, the *Cts1* gene is regulated by the MOR/RAM pathway during cell separation [22,23]. However, our results indicate that the disruption of these pathways in M. acridum had no effect on the induction of microcycle conidiation, suggesting that *MaCts1* may not be regulated by the MOR/RAM pathway during microcycle conidiation in *M. acridum*.

The efficiency of conidial germination is key for the use of entomopathogenic fungi as biological control agents [43]. In this study, the deletion of *MaCts1* resulted in delayed conidial germination. Similar phenomena were observed after the disruption of *Cts1* homologs in *Aspergillus fumigatus* [44] and *C. albicans* [25], while *Cts1* deletions had no effect on conidial germination in *Trichoderma harzianum* [45], *T. roseum* [46], *Talaromyces flavus* [47] and *Fusarium chlamydosporum* [48], indicating that *Cts1* plays distinct roles in conidial germination in different fungal species.

Conidial resistances to high temperature and UV irradiation directly determine their efficiency as mycoinsecticides in the field [49]. The fungal cell wall plays an important protective role against external stress [17,19] and chitin is one of the major cell wall structural components imparting matrix rigidity and wall maintenance [19]. Chitin synthases are essential for cell wall synthesis, and the deletion of their genes can lead to reduced cell wall integrity and tolerances to stress [17,19]. Chitinases are involved in fungal cell wall decomposition for nutrition, but can also be targeted during development, such as in mother-daughter cell separation during conidial cytokinesis [40]. In this work, deletion of *MaCts1* reduced the fungal resistances to UV-B irradiation, as well as heat-shock, which may be due to the alterations of cell wall structure and composition in Δ*MaCts1*.

In *M. acridum*, the microcycle conidiation produces higher yields of conidia with higher quality than those from typical conidiation [13]. In microcycle conidiation, the new conidia are directly separated from the germinated conidia, while in typical conidiation, multicellular mycelia form and extend from the germinated conidia [8,9]. The shift to microcycle conidiation is ultimately regulated by the availability of nutrients and other stress conditions [9]. However, the differential regulation of cell separation during microcycle vs. typical conidiation is an early and essential step in this process. Cts1 has been shown to affect the separation between mother cells and daughter cells in S. cerevisiae, leading to the formation of pseudo hyphae [21,22]. In this work, disruption of *MaCts1* led to the maintenance of typical conidiation on SYA medium, indicating that the availability of *MaCts1* plays a pivotal role in the shift to microcycle conidiation in *M. acridum* [13].

In fungi, the morphogenesis-related NDR kinase network (MOR pathway) and the regulation of Ace2 and morphogenesis (RAM) pathway in baker’s yeast are involved in morphogenetic regulation during conidiation [26]. The MOR/RAM pathways have very similar components, such as *Hym1*, *Kic1*, *Tao3*, *Cbk1* and *Mob2*, but the RAM pathway involves the zinc finger transcription factor Ace2, which is not conserved in fungi and specific for budding yeast [29,50,51]. Previous studies have shown that the expression of *Cts1* depends on genes in the RAM network, including *Tao3* and *Ace2* [22,23,52]. Both *Cts1* and *Ace2* are essential to degrade the septum for successfully detaching cells in *S. cerevisiae*, and their mutants lead to consistent defects in cell separation after mitosis [22,53]. The mutants of *Cts1* and other genes in the MOR/RAM pathway, such as *Tao3*, *Hym1* and *Kic1*, also showed similar sensitivities to oxidative stress [54]. Moreover, in other fungi, the mutants of *Cbk1* and *Ace2* homologs, essential components in the MOR/RAM pathway, displayed altered colonial morphology, defective cell separation and a decreased expression of *CTS1* [30,52,55]. We therefore initially speculated that *MaCts1* contributes to the microcycle conidiation and that it is governed by the MOR/RAM pathway in *M. acridum*. Consequently, we identified the putatively conserved *M. acridum* homologs of the MOR/RAM pathway. However, deletion of these genes did not impair the ability of *M. acridum* to induce the microcycle conidiation on SYA medium. Thus, MaCts1 contributes to the microcycle conidiation, which may not be dependent on the MOR/RAM pathway in *M. acridum*. The identification of regulatory elements in *MaCts1* expression is of interest, since this would provide insight into the network of genes governing early steps in the shift to microcycle conidiation in *M. acridum*. Future work will be directed towards predicted binding sites in the *MaCts1* promoter in combination with the analysis of differentially expressed genes in WT vs. Δ*MaCts1* during conidiation on SYA media. These results would be helpful to further clarify the molecular mechanisms underlying the microcycle conidiation in *M. acridum* and other entomopathogenic fungi.

## 5. Conclusions

The endochitinase, *MaCts1*, has been shown to make contributions to the conidial germination, conidiation yield and the resistance to UV-irradiation and heat-shock in *M. acridum*. Interestingly, *MaCts1* also contributes to the microcycle conidiation, which may be not dependent on the MOR/RAM pathway in *M. acridum*.

## Figures and Tables

**Figure 1 biology-11-01730-f001:**
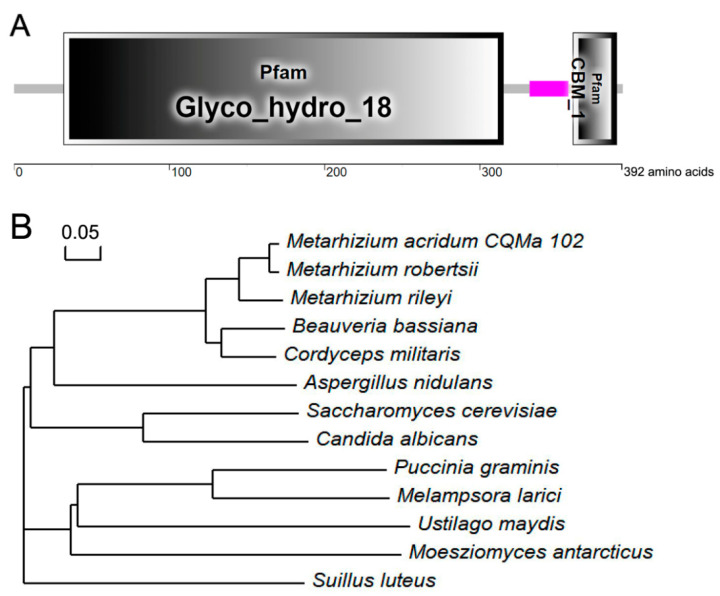
Structural and phylogenetic features of MaCts1. (**A**) Domain structure analysis of MaCts1. Glyco_hydro_18 is a glycosyl hydrolase family 18 domain and CBM_1 is a cellulose binding domain. (**B**) Phylogenetic analysis of Cts1 protein sequences from different fungi. The sequences used were *Metarhizium acridum* CQMa102, XP_007814832.1; *Metarhizium robertsii*, EXU99126.1; *Metarhizium rileyi*, TWU71928.1; *Beauveria bassiana*, KAH8712633.1; *Cordyceps militaris*, ATY58486.1; *Aspergillus nidulans*, CBF74135.1; *Saccharomyces cerevisiae*, NP_013388.1; *Candida albicans*, EAL00460.1; *Puccinia graminis*, XP_003327072.1; *Melampsora larici*, XP_007410264.1; *Ustilago maydis*, XP_011390771.1; *Moesziomyces antarcticus*, XP_014654591.1, *Suillus luteus* (KIK43475.1).

**Figure 2 biology-11-01730-f002:**
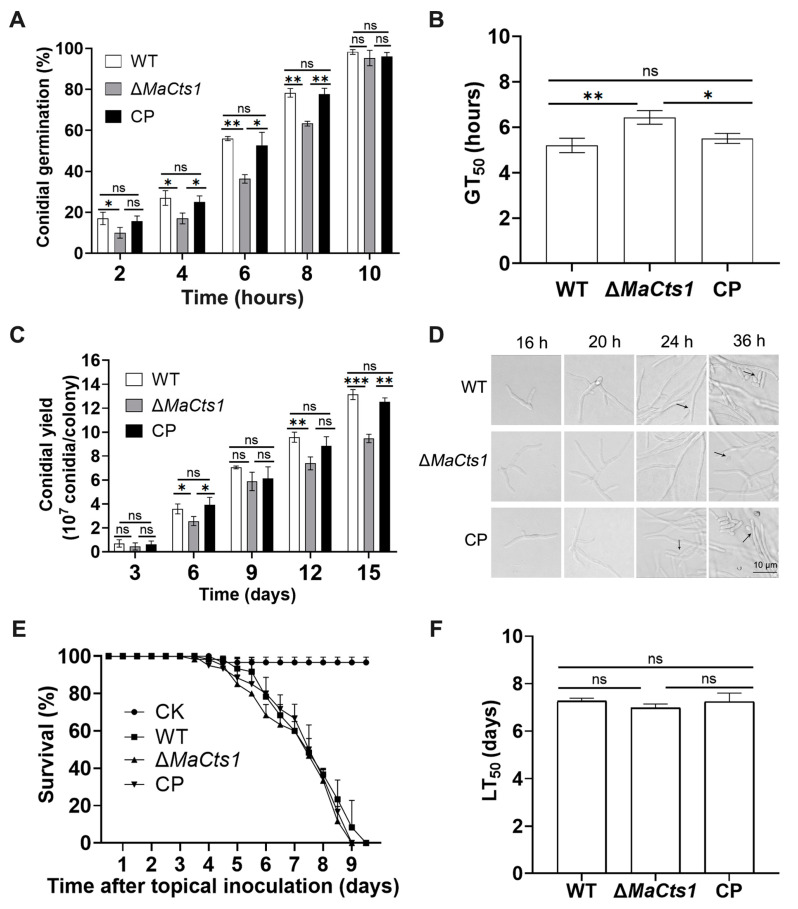
Inactivation of *MaCts1* impaired conidial germination and influenced conidial yields, but not virulence of *M. acridum*. (**A**) The conidial germination (%) of the WT, Δ*MaCts1* and CP strains examined on 1/4 SDAY medium. (**B**) The median germination time (GT_50_). (**C**) The conidial yields of the WT, Δ*MaCts1* and CP strains assessed on 1/4 SDAY medium. (**D**) The initial stage of conidiation of the WT, Δ*MaCts1* and CP strains on 1/4 SDAY medium. *Arrows* indicate conidia on the conidiophores. (**E**) Locust survival (%) after topical inoculation of conidia from the different strains. (**F**) The median lethal times (LT_50_s). *Error bars* indicate the standard deviation from triplicate experiments. One, two and three asterisks indicate a significant difference at *p* < 0.05, *p* < 0.01, *p* < 0.001, respectively. “ns” indicates no significant difference.

**Figure 3 biology-11-01730-f003:**
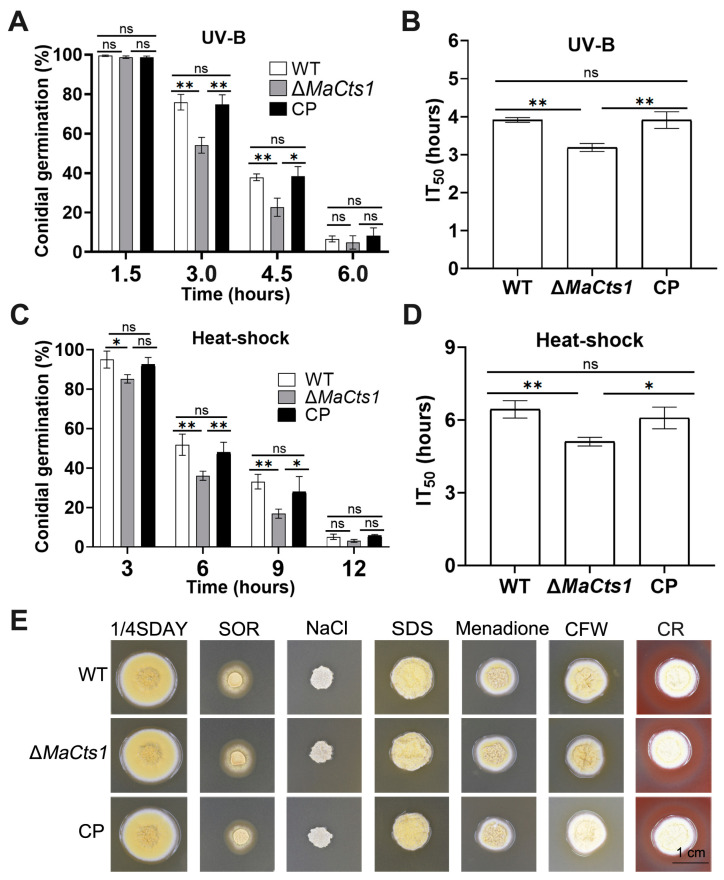
Stress tolerance assays of the WT, Δ*MaCts1* and CP strains. (**A**) The effect of UV-B irradiation on conidial germination (%) after UV-B irradiation followed by 20 h of cultivation on 1/4 SDAY medium at 28 °C. (**B**) The median inhibition times (IT_50_s) of fungal strains treated by UV-B irradiation. (**C**) The effect of heat shock (44.5 °C) on conidial germination. (**D**) IT_50_s of fungal strains treated by heat-shock (44.5 °C). (**E**) Fungal colonies on 1/4 SDAY alone or supplemented with sorbitol (SOR; 1 mol/L), calcofluor white (CFW;50 μg/mL), SDS (0.01% *w*/*v*), NaCl (1 mol/L), congo red (CR; 500 μg/mL), menadione (0.07% *w*/*v*) and SDS (0.01% *w*/*v*), respectively. *Error bars* indicate the standard deviation from triplicate experiments. One and two asterisks indicate a significant difference at *p* < 0.05 and *p* < 0.01, respectively. “ns” indicates no significant difference.

**Figure 4 biology-11-01730-f004:**
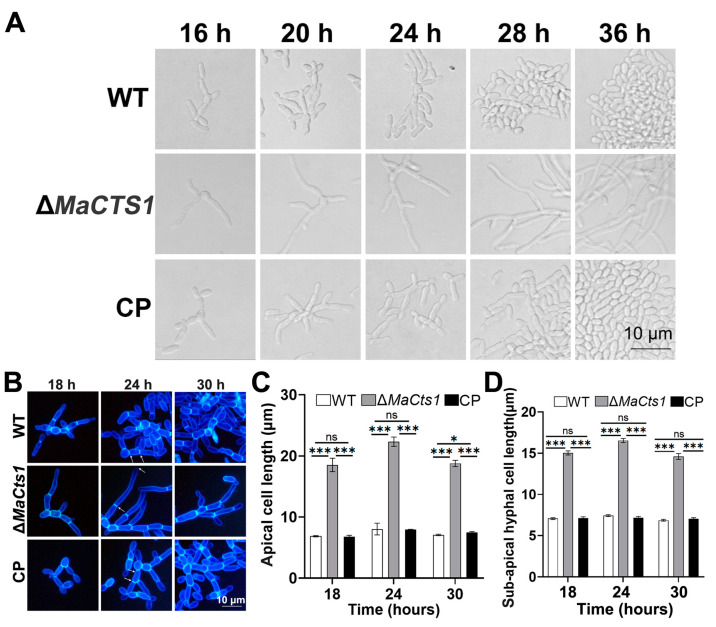
The conidiation pattern and hyphal morphology. (**A**) The conidiation pattern of the WT, Δ*MaCts1* and CP strains on SYA medium. (**B**) The hyphae from the WT, Δ*MaCts1* and CP strains grown on SYA medium for 18, 24, 30 h. The hyphae were stained with calcofluor white. *Arrows* indicate mycelial septa. (**C**) The length of apical hyphal cells in the WT, Δ*MaCts1* and CP strains. (**D**) The length of sub-apical hyphal cells in the WT, Δ*MaCts1* and CP strains. *Error bars* indicate standard deviations from triplicate experiments. Three asterisks indicate significant differences at *p* < 0.001, one asterisks indicate significant differences at *p* < 0.05 and “ns” indicates no difference.

**Figure 5 biology-11-01730-f005:**
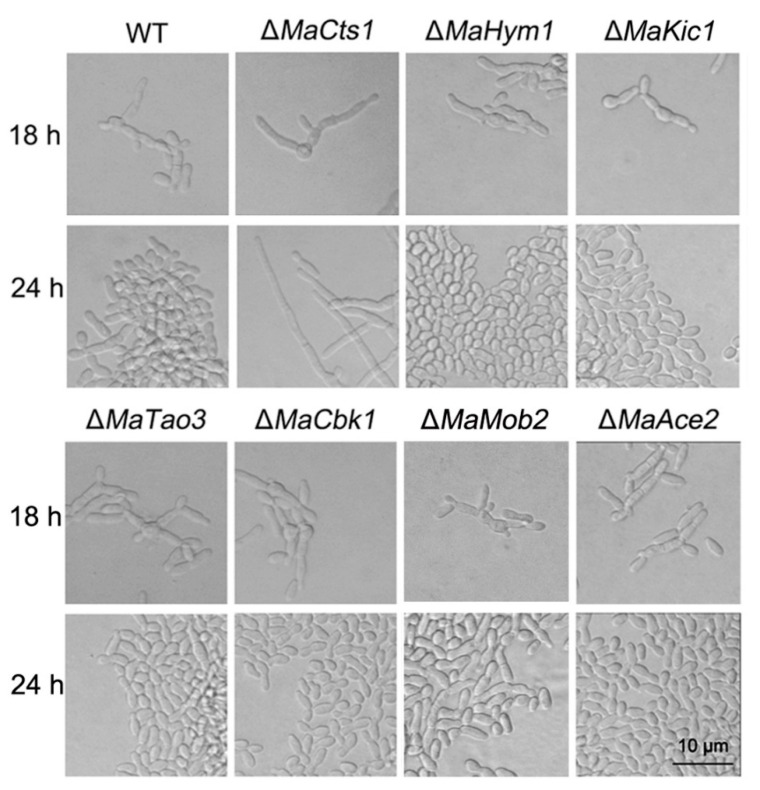
The key components of the MOR/RAM pathway are not related with the shift in conidiation pattern in *M. acridum*. The conidiation pattern of WT and disrupted mutants at 18 h and 24 h after incubation on SYA medium.

## Data Availability

Not applicable.

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
