# Peer review of "MaCts1, an Endochitinase, Is Involved in Conidial Germination, Conidial Yield, Stress Tolerances and Microcycle Conidiation in *Metarhizium acridum"

_biology, 2022, doi:10.3390/biology11121730_

Round 1

Reviewer 1 Report (New Reviewer)

The MS no. biology-2027680 (The connection of an endochitinase MaCts1 to the biocontrol potential of the entomopathogenic fungus Metarhizium acridum) by Zou and colleagues seems to be very interesting, highly effective, and well within the Journal scope. The paper is well-organized, figures are clear, understandable, and visible with even 2022 references included. In the MS, the authors focused on characterizing endochitinase gene MaCts1 in the model entomopathogenic fungus Metarhizium acridum. I detected a few flaws in the submitted paper which are described below. 

1.      As per Turnitin, MS showed 21% plagiarism when checked for similarity. I am attaching the report, please reduce.

2.      Please change the title.

3.      Please adhere to the guideline and couple Simple Summary with Abstract and present it as a single entity under Abstract.

4.      Please make sure Metarhizium acridum and gene names are italicized throughout.

5.      There is a high scope of concising the MS as well as removing the errors, I have given my edits in the Introduction section, please do in the other sections accordingly.

6.      Moreover, please italicize all the scientific names in the Reference section.

7.      Please insert the Abbreviation section after Conflict of Interests.

8.      A basic skeleton diagram should be added in the M&M section clearly pointing toward the entire methodology of the paper. This addition will definitely impact the MS influence. For reference see, http://sciencemission.com/site/index.php?page=news&type=view&id=microbiology-virology%2Fdetecting-gut-microbes&filter=8%2C9%2C10%2C11%2C12%2C13%2C14%2C16%2C17%2C18%2C19%2C20%2C27.

 I surely feel the authors can work on the above-mentioned comments easily and resubmit the Improved MS to Biology Journal. So, at present, I suggest “Accept after Minor Revisions”.

Author Response

Reviewer 2 Report (New Reviewer)

Dear authors,

I reviewed your manuscript "The connection of an endochitinase MaCts1 to the biocontrol potential of the entomopathogenic fungus Metarhizium acridum".

The study is well conducted and the manuscript is correct.  However, I consider that the manuscript needs further improvements before being published.

The English language is fine but there are minor mistakes that could be improved.  I suggest that the manuscript be read by a native speaker to edit the English language. Otherwise, the online free editor for English, "Grammarly", will help a lot.

In the abstract, the last paragraphs from lines 37 to 43 are quite difficult to understand. All short names in the abstract need to be clearly identified. I suggest simplifying these sentences, where NDR kinase network, MOR and RAM pathways are not explained, and just mention the key points or significant information.

I am not totally convinced about the chosen way to illustrate significant differences among the mutants, complementary transformants and WT strains.  The lines above the plot bars do not show the differences between all of them.  It seems that in some studies there are also differences between the deleted mutant and complementary transformants but these are not indicated.  The asterisks corresponding to a certain p-value are not indicated in all cases.  I suggest changing for the traditional letters above all bars (a, b, c, etc) to indicate significant differences.  Also, indicate clearly in the legend which p-value was used.

The additional files: "supplementary files", and "original images" have identical information.

Further corrections and suggestions are indicated in the attached file. I highlighted in red when further information or modification is necessary and in general, I added a comment.

All the best,

Author Response

Reviewer 3 Report (New Reviewer)

Author Response

This manuscript is a resubmission of an earlier submission. The following is a list of the peer review reports and author responses from that submission.

Round 1

Reviewer 1 Report

I am sorry, but the authors have failed to appropriately address the key concern previously raised concerning the various mutants of the MOR/RAM pathway. One cannot provide some experimental results that supposedly support an important conclusion (linkage or lack of linkage to the pathway) and then only vaguely analyze and discuss their significance (which at least in some way by far exceeds those of the outcome of impairing MaCts1. The authors are consistently adhering to a comparison with the yeast models even though they, themselves, state that this is not a good comparison. As Metarhizium acridum is a filamentous ascomycete, the results should have been properly compared (not just briefly cited to satisfy a reviewer) to other such fungi, especially as there are clear differences between the mentioned KOS and those described in other filamentous species (in most filamentous species the KO phenotypes are fairly similar).

In summary, the analysis of the MOR/RAM pathway genes is a very important part of this paper and, as mentioned, this part of the manuscript is still significantly flawed in spite of several rounds of resubmission. It is my opinion that without it, the other results, as solid and properly reported as they are, do not warrant sufficient novelty or progress to warrant publication in JoF.  

Reviewer 2 Report

I would like to accept the revised version of the MS.